# Using the Himawari-8 AHI Multi-Channel to Improve the Calculation Accuracy of Outgoing Longwave Radiation at the Top of the Atmosphere

**Bu-Yo Kim** [1] and **Kyu-Tae Lee** [1,2,*]

1   Research Institute for Radiation-Satellite, Gangneung-Wonju National University (GWNU), Gangneung, 7, Jukheon-gil, Gangneung, Gangwon 25457, Korea; kimbuyo@gwnu.ac.kr
2   Department of Atmospheric and Environmental Sciences, Gangneung-Wonju National University (GWNU), 7, Jukheon-gil, Gangneung, Gangwon 25457, Korea
*   Correspondence: ktlee@gwnu.ac.kr; Tel.: +82-33-640-2324

**Abstract:** In this study, Himawari-8 Advanced Himawari Imager (AHI) longwave channel data that is sensitive to clouds and absorption gas were used to improve the accuracy of the algorithm used to calculate outgoing longwave radiation (OLR) at the top of the atmosphere. A radiative transfer model with a variety of atmospheric conditions was run using Garand vertical profile data as input data. The results of the simulation showed that changes in AHI channels 8, 12, 15, and 16, which were used to calculate OLR, were sensitive to changes in cloud characteristics (cloud optical thickness and cloud height) and absorption gases (water vapor, $O_3$, $CO_2$, aerosol optical thickness) in the atmosphere. When compared to long-term analysis OLR data from 2017, as recorded by the Cloud and Earth's Radiant Energy System (CERES), the OLR calculated in this study had an annual mean bias of 2.28 $Wm^{-2}$ and a root mean square error (RMSE) of 11.03 $Wm^{-2}$. The new calculation method mitigated the problem of overestimations in OLR in mostly cloudy and overcast regions and underestimated OLR in cloud-free desert regions. It is also an improvement over the result from the existing OLR calculation algorithm, which uses window and water vapor channels.

**Keywords:** Himawari-8 Advanced Himawari Imager (AHI); multi-channel; outgoing longwave radiation at the top of the atmosphere (TOA OLR); radiative transfer model; algorithm improvement; Cloud and Earth's Radiant Energy System (CERES)

## 1. Introduction

Outgoing longwave radiation at the top of the atmosphere (TOA OLR) is an indicator that can describe the overall state of the earth-atmosphere system [1–4]. Also, OLR is an important radiation budget when balanced with net shortwave radiation at the top of the atmosphere and used in climate studies related to energy balance [5–10]. OLR values change continually due to changes in surface temperature, the atmosphere, and clouds [11]. As changes in OLR are sensitive to the temperature emissions from land, ocean, and clouds, it has been used actively in several studies [12]. The lower OLR emitted from cloudy areas (i.e., 240 $Wm^{-2}$ or less) is closely connected to convective activity and is very useful in the observation of tropical cyclones [13]. Changes in OLR also have a strong correlation with large-scale convection systems [10] such as El Niño [14], La Niña [15], the El Niño Southern Oscillation (ENSO) [16], and the Madden–Julian Oscillation (MJO) [17,18]. There is also a relationship between sea surface temperature (SST) and OLR, because medium-scale weather changes that occur over the ocean and in the atmosphere are very important for predicting monsoon periods [19]. Thus, OLR is also used in studies on predicting rainy seasons and rainfall amounts in equatorial regions where yearly

rainfall is high, such as India (the Indian Ocean), Brazil, and parts of Africa [20–23]. Furthermore, daily changes in OLR can quantitatively show the state of the weather, including surface temperature, cloud characteristics, and rainfall amounts, because it reflects overall changes in the earth–atmosphere system; it is also used to study a variety of weather phenomena [11,24,25]. In addition, OLR is also being used to predict typhoons and earthquakes based on differences in OLR emitted based on changes in surface temperature and cloud conditions [22,26,27].

The production and continuous monitoring of reliable, high-quality OLR data is very important for predictions of future climate change [28]. The OLR observed by broadband sensors on polar satellites (wavelength regions of 3–100 μm) is highly accurate, but its low spatiotemporal resolution is a drawback. Therefore, many past and current studies have been conducted using narrowband sensors on geostationary satellites that have a high spatiotemporal resolution [4]. The initial algorithm developed used a single window channel's brightness temperature [29–31]; however, OLR calculated from this kind of data does not reflect varied information on the state of the atmosphere. Therefore, researchers then developed multi-spectral algorithms that included water vapor or other window channels [2,4,32,33]. These studies were based on the high correlation between the radiation measured by narrowband and broadband sensors [34]. Goldberg et al. [35], Doelling et al. [36,37], and Kim and Lee [38] performed studies on reducing the empirical calculation error for OLR calculated from narrowband sensors.

There is a large difference between the OLR emitted from the earth's surface (land and ocean) and from cloudy areas; therefore, it is possible to estimate this parameter using only a single infrared channel that can describe this difference well. However, to analyze and predict the radiation budget, weather, and climate change, OLR must react sensitively to absorption gases in the atmosphere in clear sky (cloud-free) condition. Water vapor is already known as a very important factor in reducing OLR [10,39]. $CO_2$ also reduces OLR as its concentrations increase continually [40], and there are also studies on OLR reductions caused by $O_3$ and aerosols [41–44]. However, these must use channels that are sensitive to absorption gases because there are regional patterns and long-term changes in this factor and increases in the concentration of absorption gases in the atmosphere reduce OLR [15]. The present study is an advanced study on the development of the radiation calculation algorithm used by the Geostationary Korea Multi-Purpose satellite 2A (GK-2A). It improves OLR calculation by adding channels that are sensitive to $O_3$ and $CO_2$ to the algorithm developed by Kim et al. [4], which is based on the window and water vapor channels that are part of the Himawari-8 Advanced Himawari Imager (AHI). The results of the new OLR algorithm were then compared to that from the Cloud and Earth's Radiant Energy System (CERES) OLR [45,46].

## 2. Research Data and Methodology

In this study, sensor data from the Himawari-8 AHI geostationary satellite was used to calculate OLR. The sensors consist of 6 shortwave channels and 10 longwave narrowband channels. A hemispheric region including the Pacific Ocean (with central coordinates of 0°N, 140.7°E) was observed at a spatial resolution of 2 km × 2 km and a temporal resolution of 10-min [47,48]. The longwave channel-specific characteristics of the AHI used in this study are shown in Table 1 [49,50]. The observed longwave channel data was converted into radiance and used to calculate the OLR. In this process, it was necessary to include a process that converted narrowband radiance into narrowband irradiance and one that converted narrowband irradiance into broadband irradiance (that is OLR; see Section 3). The OLR calculated in this study, hereafter "improved OLR algorithm", was compared with that developed by Kim et al. [4] and CERES.

**Table 1.** Himawari-8 AHI channel-specific central wavelengths and primary purpose.

| Channel | Descriptive Name | Central Wavelength [μm] | Primary Purpose |
|---------|------------------|-------------------------|-----------------|
| 7 | Shortwave window | 3.89 | Surface and cloud, fog at night, fire, and winds |
| 8 | Upper-level water vapor | 6.24 | High-level atmospheric water vapor, winds, and rainfall |
| 9 | Mid-level water vapor | 6.94 | Mid-level atmospheric water vapor, winds, and rainfall |
| 10 | Lower-level/Mid-level water vapor | 7.35 | Lower-level atmospheric water vapor, winds, and $SO_2$ |
| 11 | Cloud-top phase | 8.59 | Total water for stability, cloud phase, dust, $SO_2$, and rainfall |
| 12 | $O_3$ | 9.64 | Total ozone, turbulence, and winds |
| 13 | Clean longwave window | 10.41 | Surface and cloud |
| 14 | Longwave window | 11.24 | Imagery, sea surface temperature, clouds, and rainfall |
| 15 | Dirty longwave window | 12.38 | Total water, ash, and sea surface temperature |
| 16 | $CO_2$ | 13.28 | Air temperature, cloud heights and amounts |

The CERES OLR used in the comparative analysis came from the CERES Single Scanner Footprint (SSF) XTRK Edition4A installed on the Terra polar satellite. This data was constantly observed at a spatial resolution of 20 km × 20 km within CERES' field of view. CERES provides radiation data calculated from highly accurate cloud detections and it is more accurate than other radiation data [9,27,51–53]. However, CERES observations use a different spatiotemporal resolution than AHI does; therefore, spatiotemporal resolution matching must be performed to compare the two OLR values. Ellingson et al. [54], Ba et al. [6], and Park et al. [55] performed a comparative analysis that assumed that the state of the atmosphere did not change within 30-min and averaged long-term data collected at a spatial resolution of 1° × 1°. However, in this case, the root mean square error (RMSE) of the standard data decreased as the averaged area became larger and the quantity of data increased [12]. Furthermore, when high-resolution data is compared to low-resolution data, the difference in the radiation observed over cloud-free areas is not large; however, there is a large difference in cloudy areas, especially at the edge of the clouds. As such, the comparative analysis must be performed at a finer spatiotemporal resolution [4,56].

This study followed the method performed by Kim et al. [4] and averaged the OLR from the AHI in a 20 km × 20 km area based on the coordinates and spatial resolution of the observed CERES data. The comparative analysis used the observed CERES OLR within ±5-min of the AHI observation. It also only used data that corresponded to an area with a viewing zenith angle (VZA) of less than 70°, following the central coordinates of the AHI. In the example in Figure 1, the sample of AHI daytime OLR created at 0100 UTC was compared with the CERES OLR observed at 0055–0105 UTC. The data for 0110, 0120, and 0130 UTC was compared with that for 0105–0115, 0115–0125, and 0125–0135 UTC. Similarly, AHI nighttime OLR observed at 1320, 1330, 1340, and 1350 UTC was compared to the CERES OLR observed from 1315–1325, 1325–1335, 1335–1345, and 1345–1355 UTC, respectively.

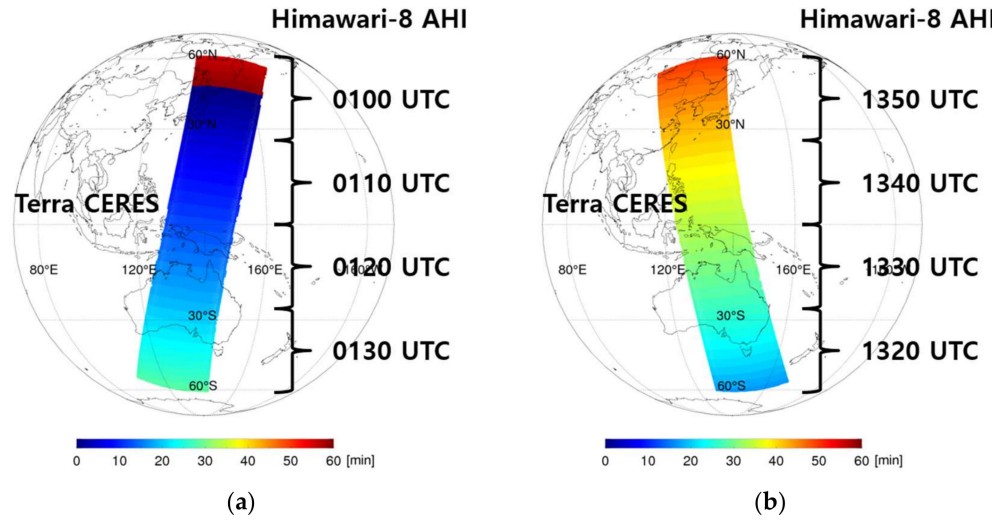

**Figure 1.** Daytime (**a**) and nighttime (**b**) examples of observation area according to observation time of Terra CERES (color bar) and Himawari-8 AHI.

The comparative analysis used daytime and nighttime data from CERES as it passed over the same observation area for a period of about 16 days between January and December 2017 (Table 2). The mean, bias, RMSE, and correlation coefficient of the OLR calculated in this study were analyzed and OLR was compared to CERES. It was also compared with the OLR originally developed by Kim et al. [4], particularly in terms of the differences resulting from the improvements made to the OLR in this study, including the improved accuracy. There was a large difference in the OLR given the presence and amount of clouds; therefore, the analysis was divided into cloud-free and cloudy areas. Following CERES' clear sky fraction (0–100%), the cloud-free area was set at 95–100% and the cloudy area was categorized as partly cloudy (50–95%), mostly cloudy (5–50%), or overcast (0–5%). For the cloud-free area, the surface types used in CERES (1–20) were followed; the areas were categorized as ocean (17, 20) and land (1–16, 18, 19) [57].

**Table 2.** Observation time used in the comparative analysis of CERES and AHI.

| Daytime | | | Nighttime | | |
|---|---|---|---|---|---|
| **Date (Month/Day)** | | **Time (UTC)** | **Date (Month/Day)** | | **Time (UTC)** |
| 01/04 | 01/20 | | 01/04 | - | |
| 02/05 | 02/21 | | 02/05 | 02/21 | |
| 03/09 | 03/25 | | 03/09 | 03/25 | |
| 04/10 | 04/26 | | 04/10 | 04/26 | |
| 05/12 | 05/28 | | 05/12 | 05/28 | |
| 06/13 | 06/29 | 0100, 0110, | 06/13 | 06/29 * | 1320, 1330, |
| 07/15 * | 07/31 | 0120, 0130 | 07/15 | 07/31 | 1340, 1350 |
| 08/07 | 08/23 | | 08/07 | 08/23 | |
| 09/08 | 09/29 | | 09/08 | 09/29 | |
| 10/10 | 10/26 | | 10/10 * | 10/26 * | |
| 11/11 | 11/27 | | 11/11 * | 11/27 | |
| 12/13 * | 12/29 | | 12/13 * | 12/29 * | |

* The daytime observation data from 07/15 0100 UTC and 12/13 0130 UTC, and the nighttime observation data from 06/29 1330–1350 UTC, 10/10 1340 UTC, 10/26 1320 UTC, 11/11 1320 UTC, 12/13 1330, 1350 UTC, and 12/29 1340 UTC were excluded from the analysis of the AHI observation data. - Excluded from analysis because they were not present in the CERES data.

## 3. TOA OLR Calculation Algorithm

Figure 2 shows a flowchart of the algorithm used to calculate OLR. As this study used the AHI's narrowband channel data for this task, it was necessary to include a process that converted the

observed narrowband radiance of each channel into broadband irradiance (OLR; see Figure 2b) [58]. During this conversion process, the OLR was calculated using the regression coefficient based on the linear relationship between narrowband and broadband radiation calculated for various atmospheric conditions using the Santa Barbara DISORT Atmospheric Radiative Transfer (SBDART) model [59] (see Section 3.1). SBDART has been used in many studies because it has high accuracy with less than 3% error compared to the shortwave/longwave radiation spectrum measured by Atmospheric Emitted Radiance Interferometer (AERI) [60,61], Precision Spectral Pyranometer (PSP), and Normal Incidence Pyrheliometer (NIP) [59], and it calculates various atmospheric conditions very quickly [3,4,38,55,62,63]. The relationship between the narrowband radiance and irradiance of each channel simulated in SBDART was used to create a regression coefficient that converted the radiance to irradiance (see Section 3.2). Finally, the relationship between the narrowband irradiance and the 3.3–100 μm wavelength region's broadband irradiance (OLR) of each channel was used to create the regression coefficient that converted the narrowband irradiance of each channel to OLR (see Section 3.3).

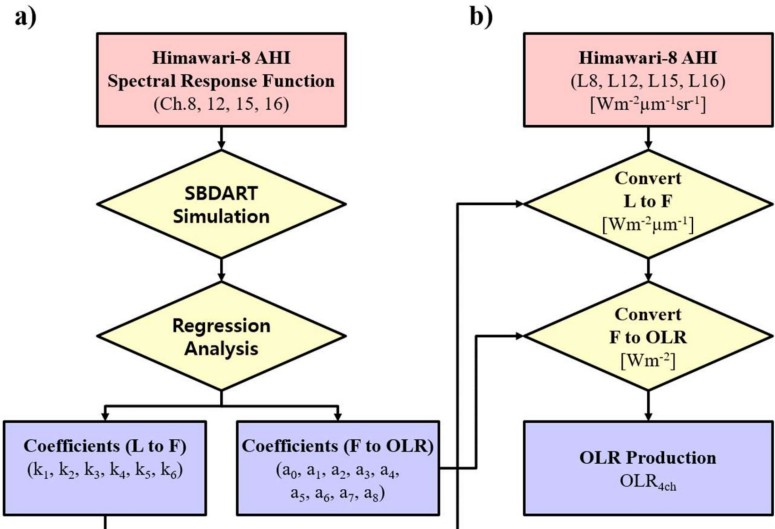

**Figure 2.** Flowchart of the algorithm used to calculate OLR using Himawari-8 AHI channel data and the algorithms used to calculate the regression coefficients (**a**) needed to convert between different types of radiation at different stages (**b**). L is the narrowband radiance and F is the narrowband irradiance.

### 3.1. Radiative Transfer Model Sensitivity Test

There is a large difference in the characteristics of TOA OLR emitted from the surface of the earth and from cloud. In cloud-free areas, the OLR is reduced by absorption gases in the atmosphere [15]. Therefore, it is necessary to simulate a variety of atmospheric conditions when using a radiative transfer model to estimate OLR [3,4,11,64]. When narrowband channel data such as that from AHI is used, the accuracy of the OLR produced can vary according to the channel used. A single channel algorithm that used window channel data of approximately 12.4 μm was able to describe approximately 97% of changes in OLR, but it was not sensitive in terms of reflecting reductions caused by absorption gases such as $O_3$ or $CO_2$. Channels in the vicinity of 6.9 μm, 9.6 μm, and 13.3 μm, however, do include changes in OLR related to absorption gases in the atmosphere, and using these made it possible to improve the accuracy of the OLR calculation [6,10,54]. As the number of channel utilized increased, the accuracy improved; but when channels that had similar features around certain wavelengths were used, the improvement was not significant and similar trends were seen [64]. Therefore, in this study, tests of sensitivity for cloud optical thickness (COT) and absorption gases were performed as shown in Figure 3. AHI channels sensitive to $O_3$ and $CO_2$ were added to the algorithm developed by Kim et al. [4], which uses window and water vapor channels, as improvements.

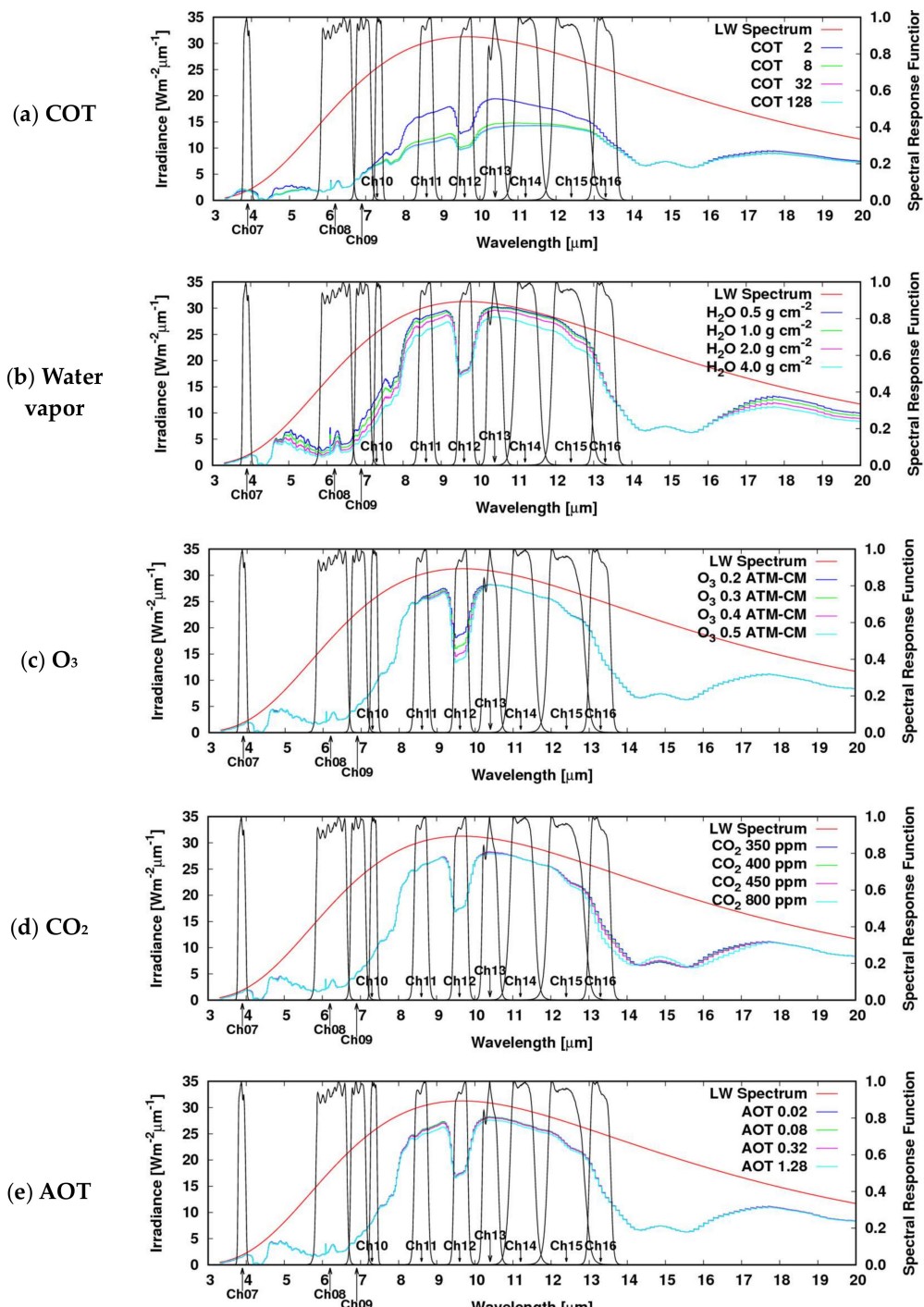

**Figure 3.** Changes in OLR for different wavelengths based on (**a**) COT (blue: 2, green: 8, magenta: 32, and cyan: 128), concentration of the following absorption gases in the atmosphere (**b**) water vapor (blue: 0.5 g·cm$^{-2}$, green: 1.0 g·cm$^{-2}$, magenta: 2.0 g·cm$^{-2}$, and cyan: 4.0 g·cm$^{-2}$), (**c**) O$_3$ (blue: 0.2 ATM-CM, green: 0.3 ATM-CM, magenta: 0.4 ATM-CM, and cyan: 0.5 ATM-CM), and (**d**) CO$_2$ (blue: 350 ppm, green: 400 ppm, magenta: 450 ppm, and cyan: 800 ppm), and (**e**) AOT (blue: 0.02, green: 0.08, magenta: 0.32, and cyan: 1.28). The red line is the longwave radiation spectrum and the black line is the spectral response function for each channel of the Himawari-8 AHI.

Figure 3 shows the changes in OLR at each wavelength according to the COT and aerosol optical thickness (AOT) and concentration of absorption gases (water vapor, O$_3$, CO$_2$) in the 3.3–20 μm wavelength region. The vertical profile used as input data in the radiative transfer model for the

sensitivity tests was set to tropical. The characteristics of the aerosols and the cloud (with cloud ceiling height set to 6 km) were input only in the COT and AOT sensitivity tests. In the AOT sensitivity test, rural, urban, oceanic, and tropospheric were entered as aerosol characteristics in the radiative transfer model, and the calculated results were averaged as shown in Figure 3 and Table 3. The $CO_2$ concentration was set at 400 ppm in the same way except in the $CO_2$ sensitivity test. The sensitivity test results in Table 3 show the OLR changes integrated within the spectral response function of a channel according to the COT and AOT and concentration of the absorption gases in each AHI channel. OLR increases in proportion to the earth's surface and the atmospheric temperature in the overall longwave area. However, when clouds were present, the OLR in the window channel decreased significantly and the OLR decreased due to an increase in COT (Figure 3a). Water vapor is the absorption gas that has the greatest effect on OLR reduction, and sensitive changes were seen in both the window and water vapor channels. The wavelength region was large in channel 15 and the OLR here was larger than in other channels. A large reduction of more than 2.7 $Wm^{-2}$ was also seen following changes in COT and water vapor. Channels 13 and 14 showed similar characteristics, but their OLR was smaller than that in channel 15 and they were not as sensitive to water vapor. Similarly, channel 8 (the water vapor channel) showed the largest OLR and large changes in OLR according to water vapor. In contrast, $O_3$ and $CO_2$ in channels 12 and 16 showed clear differences from other channels near the 9.7 μm and 13.3 μm wavelengths. When $CO_2$ was assumed to have a concentration of 800 ppm, which is twice the assumed concentration of 400 ppm, OLR was reduced by approximately 1 $Wm^{-2}$. The global mean concentration of $CO_2$ is increasing continually and must be taken into account because, like water vapor, it is a major absorption gas contributing to weather and climate change related to global warming [10,39]. The changes in OLR were not clear, in terms of AOT, compared to other absorption gases; however, a decreasing OLR trend was seen in the window channel. There was a 0.64 $Wm^{-2}$ change in channel 15, which had the largest wavelength region, due to AOT. Urban aerosols showed the largest change, at 1.09 $Wm^{-2}$, in their category; rural aerosols showed a small change of 0.39 $Wm^{-2}$. This shows that channel 15 can reflect OLR changes not only due to COT and water vapor but also due to $CO_2$ and AOT. However, because it is difficult to reflect various atmospheric states and reductions in absorption gases using channel 15 alone, channel 8 (water vapor) and channels 12 and 16 ($O_3$ and $CO_2$, respectively) were used with the expectation that this would improve the accuracy of OLR calculations [10,15].

This study used 42 kinds of vertical profile data from Garand et al. [65] as inputs to simulate the radiative transfer model under various atmospheric conditions. This data includes vertical profile data for six standard atmospheres (tropical, mid-latitude summer and winter, subarctic summer and winter, and U.S. standard; profiles 1–6, respectively) and a variety of changes in atmospheric temperature (profiles 7–18), amount of water vapor (19–30), and amount of ozone (31–42) [65]. Other options in the radiative transfer model are shown in Table 4. As clouds cause the largest reduction effect, the simulation was performed in detail based on cloud height and optical thickness. Water vapor and $O_3$ data were entered differently based on the vertical profiles. The $CO_2$ concentration entered was 400 ppm and default values were used for trace gases [59].

**Table 3.** Changes in OLR based on COT, AOT and concentration of absorption gases (water vapor, $O_3$, and $CO_2$) for each channel in the AHI. "Diff." is the difference between the maximum and minimum radiation of the sensitivity test for each condition.

| | | AHI Longwave Channels | | | | | | | | | |
|---|---|---|---|---|---|---|---|---|---|---|---|
| | | 7 | 8 | 9 | 10 | 11 | 12 | 13 | 14 | 15 | 16 |
| COT | 2 | 0.33 | 1.97 | 2.03 | 1.36 | 5.72 | 5.05 | 9.03 | 12.48 | 17.53 | 6.72 |
| | 8 | 0.37 | 1.96 | 1.99 | 1.25 | 4.03 | 3.95 | 6.78 | 9.96 | 15.06 | 6.19 |
| | 32 | 0.36 | 1.96 | 1.97 | 1.22 | 3.81 | 3.80 | 6.46 | 9.67 | 14.80 | 6.13 |
| | 128 | 0.36 | 1.95 | 1.96 | 1.21 | 3.78 | 3.78 | 6.42 | 9.62 | 14.75 | 6.11 |
| | **Diff.** | **−0.03** | **0.02** | **0.07** | **0.15** | **1.93** | **1.28** | **2.61** | **2.86** | **2.78** | **0.61** |
| Water vapor ($g \cdot cm^{-2}$) | 0.5 | 0.27 | 4.18 | 3.41 | 2.36 | 9.88 | 7.03 | 14.08 | 19.86 | 27.98 | 9.02 |
| | 1.0 | 0.26 | 3.53 | 2.90 | 2.07 | 9.71 | 7.01 | 14.00 | 19.73 | 27.66 | 8.97 |
| | 2.0 | 0.26 | 2.97 | 2.47 | 1.79 | 9.42 | 6.95 | 13.77 | 19.33 | 26.89 | 8.84 |
| | 4.0 | 0.25 | 2.50 | 2.10 | 1.55 | 8.89 | 6.77 | 13.17 | 18.34 | 25.29 | 8.51 |
| | **Diff.** | **0.02** | **1.68** | **1.31** | **0.81** | **0.99** | **0.27** | **0.91** | **1.52** | **2.70** | **0.51** |
| $O_3$ (ATM-CM) | 0.2 | 0.25 | 1.98 | 2.08 | 1.54 | 8.89 | 7.20 | 13.15 | 18.28 | 25.21 | 8.50 |
| | 0.3 | 0.25 | 1.98 | 2.08 | 1.54 | 8.84 | 6.43 | 13.11 | 18.28 | 25.18 | 8.49 |
| | 0.4 | 0.25 | 1.98 | 2.08 | 1.54 | 8.79 | 5.86 | 13.08 | 18.28 | 25.15 | 8.49 |
| | 0.5 | 0.25 | 1.98 | 2.08 | 1.54 | 8.75 | 5.44 | 13.05 | 18.28 | 25.12 | 8.48 |
| | **Diff.** | **0.00** | **0.00** | **0.00** | **0.00** | **0.13** | **1.77** | **0.11** | **0.00** | **0.09** | **0.02** |
| $CO_2$ (ppm) | 350 | 0.25 | 1.98 | 2.08 | 1.54 | 8.86 | 6.75 | 13.13 | 18.28 | 25.21 | 8.53 |
| | 400 | 0.25 | 1.98 | 2.08 | 1.54 | 8.86 | 6.75 | 13.12 | 18.28 | 25.15 | 8.35 |
| | 450 | 0.25 | 1.98 | 2.08 | 1.54 | 8.86 | 6.75 | 13.10 | 18.27 | 25.09 | 8.19 |
| | 800 | 0.25 | 1.98 | 2.08 | 1.54 | 8.86 | 6.74 | 13.00 | 18.25 | 24.75 | 7.35 |
| | **Diff.** | **0.00** | **0.00** | **0.00** | **0.00** | **0.00** | **0.02** | **0.14** | **0.02** | **0.46** | **1.18** |
| AOT | 0.02 | 0.16 | 1.72 | 1.82 | 1.34 | 7.06 | 4.97 | 10.75 | 15.29 | 21.64 | 7.57 |
| | 0.08 | 0.16 | 1.72 | 1.81 | 1.34 | 7.05 | 4.97 | 10.73 | 15.27 | 21.60 | 7.56 |
| | 0.32 | 0.16 | 1.72 | 1.81 | 1.33 | 6.98 | 4.95 | 10.66 | 15.18 | 21.47 | 7.53 |
| | 1.28 | 0.19 | 1.70 | 1.79 | 1.32 | 6.76 | 4.87 | 10.41 | 14.86 | 21.00 | 7.41 |
| | **Diff.** | **−0.03** | **0.02** | **0.02** | **0.02** | **0.30** | **0.10** | **0.34** | **0.44** | **0.64** | **0.16** |

**Table 4.** Radiative transfer model setting options used to calculate OLR.

| Parameter | Values | N |
|---|---|---|
| Spectral band [μm] | 5.44–7.03 (Ch.08), 9.33–9.93 (Ch.12), 11.18–13.65 (Ch.15), 12.86–13.76 (Ch.16), 3.3–100 (broadband) | 5 |
| Spectral resolution [μm] | 0.005 | |
| Atmospheric profile | Garand profiles | 42 |
| VZA [°] | 0, 5, 10, 15, 20, 25, 30, 35, 40, 45, 50, 55, 60, 65, 70, 75, 80, 85 | 18 |
| COT | 2, 4, 8, 16, 32, 64, 128 | 7 |
| Cloud height [km] | 0, 2, 4, 6, 8, 10, 12, 14, 16 | 9 |
| Surface temperature, water vapor, $O_3$, etc. | Garand profiles | |
| $CO_2$ | 400 ppm | |

## *3.2. Converting Radiance to Irradiance*

If the radiation emitted from the atmosphere was isotropic, $F = \pi L$ could be established. However, because this radiation is actually anisotropic, the radiance observed from the satellite's VZA must be converted to irradiance [1]. Therefore, the equation given below is used to convert the narrowband radiance observed on the AHI channel into narrowband irradiance (Equations (1)–(3)).

$$F = A(\theta)L(\theta) + B(\theta) \tag{1}$$

$$A(\theta) = k_1 + k_2(\sec\theta - 1) + k_3(\sec\theta - 1)^2 \tag{2}$$

$$B(\theta) = k_4 + k_5(\sec\theta - 1) + k_6(\sec\theta - 1)^2 \tag{3}$$

Here, $\theta$ is the VZA, L is the narrowband radiance ($Wm^{-2}\mu m^{-1}sr^{-1}$), F is the narrowband irradiance ($Wm^{-2}\mu m^{-1}$), $k_{1-6}$ are the L-F regression coefficients, and A and B are the empirical limb darkening functions [11,66]. The L-F regression coefficients used in this study are shown in Table 5. The narrowband irradiance, which was converted when the VZA was 70° or less, showed %RMSEs

(= RMSE/mean$\times$100%) of 0.09%, 0.20%, 0.14%, and 0.07%, respectively for the mean narrowband irradiances of 1.20 Wm$^{-2}$, 2.87 Wm$^{-2}$, 10.79 Wm$^{-2}$, and 4.65 Wm$^{-2}$ simulated for each channel in the radiative transfer model.

**Table 5.** The regression coefficient for each channel, used to convert narrowband radiance to narrowband irradiance.

| Channel | $k_1$ | $k_2$ | $k_3$ | $k_4$ | $k_5$ | $k_6$ |
|---|---|---|---|---|---|---|
| 8 | $2.670 \times 10^0$ | $7.084 \times 10^{-1}$ | $-4.046 \times 10^{-2}$ | $9.869 \times 10^{-2}$ | $-1.424 \times 10^{-1}$ | $8.770 \times 10^{-3}$ |
| 12 | $2.425 \times 10^0$ | $1.138 \times 10^0$ | $-9.878 \times 10^{-2}$ | $4.941 \times 10^{-1}$ | $-7.435 \times 10^{-1}$ | $6.409 \times 10^{-2}$ |
| 15 | $3.017 \times 10^0$ | $1.125 \times 10^{-1}$ | $-5.046 \times 10^{-3}$ | $5.252 \times 10^{-2}$ | $9.457 \times 10^{-2}$ | $-1.095 \times 10^{-2}$ |
| 16 | $2.791 \times 10^0$ | $4.747 \times 10^{-1}$ | $-1.817 \times 10^{-2}$ | $3.331 \times 10^{-1}$ | $-4.345 \times 10^{-1}$ | $1.454 \times 10^{-2}$ |

*3.3. Converting Irradiance to TOA OLR*

To convert the narrowband irradiance calculated in Section 3.2 into the broadband irradiance that is OLR, the process in Equation (4) using AHI channels 8, 12, 15, and 16 was followed.

$$\mathrm{OLR_{4ch}} = a_0 + a_1 F_{6.2} + a_2 F_{6.2}^2 + a_3 F_{9.6} + a_4 F_{9.6}^2 + a_5 \ln(F_{12.4}) + a_6 \ln(F_{12.4})^2 + a_7 F_{13.3} + a_8 F_{13.3}^2 \quad (4)$$

Here, $F_{6.2}$, $F_{9.6}$, $F_{12.4}$, and $F_{13.3}$ are the narrowband irradiances for each channel and $a_{0-8}$ are the F-OLR regression coefficients. The F-OLR regression coefficients derived are shown in Table 6.

**Table 6.** Regression coefficients for converting narrowband irradiance to OLR.

| $a_0$ | $a_1$ | $a_2$ | $a_3$ | $a_4$ | $a_5$ | $a_6$ | $a_7$ | $a_8$ |
|---|---|---|---|---|---|---|---|---|
| 90.257 | 1.474 | 0.0174 | 1.743 | 0.00213 | $-44.823$ | 28.208 | 1.612 | 0.00952 |

The OLR$_{4ch}$ that was converted to OLR, and the OLR$_{3.3-100}$ that was simulated in the radiative transfer model and integrated by 3.3–100 μm wavelength region are shown in the scatter diagram in Figure 4. The two OLR values were very similar with a correlation coefficient of 0.998, and the RMSE of 2.85 Wm$^{-2}$ resulted in a %RMSE of 1.87%. This is a smaller difference than the %RMSE of 2–2.2% between the results of the OLR developed by Schmetz and Liu [11], Clerbaux et al. [3], and Kim et al. [4] and the radiative transfer model.

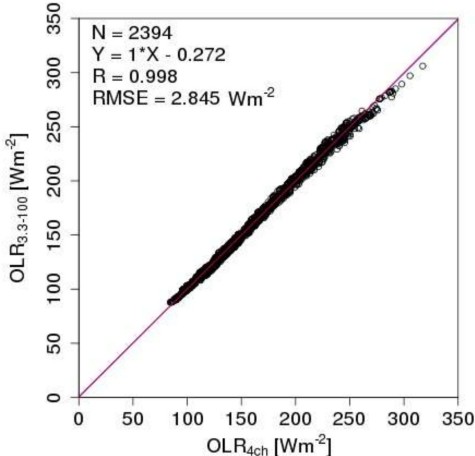

**Figure 4.** Scatter diagram of OLR$_{4ch}$ and the results of the radiative transfer model simulation integrated by 3.3–100 μm wavelength region (OLR$_{3.3-100}$). The 1:1 line is in red and the regression line is in blue.

## 4. Results

$OLR_{1ch}$, which was calculated using only the window channel as developed by Kim et al. [4], and $OLR_{2ch}$, which was calculated using the window and water vapor channels, were different from $OLR_{4ch}$, which was calculated using the improvements made in this study, as shown in Figure 5. This figure shows the difference between CERES and $OLR_{4ch}$ results using 4 January 2017, as a daytime/nighttime case. This was compared to the results of the $OLR_{1ch}$ and $OLR_{2ch}$ calculations. In the daytime/nighttime case, the OLR was larger in cloud-free areas where the temperature was relatively high compared to the cloudy areas. In the daytime case, the largest OLR distribution was around the Australian desert. Here, the bias between $OLR_{4ch}$ and CERES OLR was 2.38 $Wm^{-2}$, which was a difference of 0.99% against the mean OLR. The cloudy areas had a difference of 2.72 $Wm^{-2}$ (1.15%) while the cloud-free areas showed a difference of $-0.18$ $Wm^{-2}$ ($-0.07\%$). This difference was smaller than the difference with $OLR_{1ch}$ (all cases: 4.78 $Wm^{-2}$ (2.00%), cloudy cases: 5.66 $Wm^{-2}$ (2.40%), and cloud-free cases: $-1.80$ $Wm^{-2}$ ($-0.66\%$)) and $OLR_{2ch}$ (all cases: 4.79 $Wm^{-2}$ (2.00%), cloudy cases: 5.58 $Wm^{-2}$ (2.37%), and cloud-free cases: $-1.11$ $Wm^{-2}$ ($-0.41\%$)), which had a larger bias than the CERES OLR. The RMSEs (%RMSE) of $OLR_{1ch}$, $OLR_{2ch}$, and $OLR_{4ch}$ with CERES OLR were 12.38 $Wm^{-2}$ (5.16%), 11.68 $Wm^{-2}$ (4.87%), and 10.86 $Wm^{-2}$ (4.53%), respectively, indicating that $OLR_{4ch}$ had the smallest difference. The correlation coefficients of the OLR values calculated in this study and the CERES OLR were similar, ranging between 0.97 and 0.99.

The $OLR_{4ch}$ calculated had a relatively large difference (%bias of 1.43%) compared to the CERES OLR in overcast areas (partly cloudy cases: 0.76% and mostly cloudy cases: 1.06%). This difference occurred because the OLR difference was high in areas at the edge of the clouds due to errors that occurred because the spatiotemporal resolutions of the two OLR values compared was different [4,67]. The $OLR_{1ch}$ and $OLR_{2ch}$ calculation algorithms had difficulties calculating values that were smaller than the CERES OLR values in hot dry regions such as deserts in the daytime. As shown in Figure 5i,j, the improved $OLR_{4ch}$ obtained in this study ameliorated the problem of the calculated OLR being greater than the CERES OLR in cloudy regions and less than the CERES OLR in regions with a large OLR in daytime; this was a problem in the $OLR_{1ch}$ and $OLR_{2ch}$ calculations. Therefore, the bias (%bias) and RMSE (%RMSE) of $OLR_{4ch}$ and CERES OLR in the cloud-free region around the Australian desert in the daytime case were 1.14 $Wm^{-2}$ (0.35%) and 6.73 $Wm^{-2}$ (2.05%), respectively. This difference was smaller than those of $OLR_{1ch}$ ($-7.36$ $Wm^{-2}$ ($-2.24\%$) and 10.61 $Wm^{-2}$ (3.23%)) and $OLR_{2ch}$ ($-6.11$ $Wm^{-2}$ ($-1.86\%$) and 8.92 $Wm^{-2}$ (2.71%)). The Australian desert was classified according to the desert (7) and savannah (9) surface type data provided in CERES, which is based on the International Geosphere-Biosphere Programme classifications (IGBP) [57].

Figure 6 shows the results of a long-term comparative analysis of the cases selected in this study. Figure 6a shows the monthly bias and RMSEs of $OLR_{1ch}$, $OLR_{2ch}$, and $OLR_{4ch}$ for all the cases and the number of data analyzed for each month. The comparative analysis results of the annual means in Figure 6a were categorized in detail according to the clouds and earth surface characteristics in Table 7. Compared to $OLR_{1ch}$ and $OLR_{2ch}$, the monthly bias and RMSE of $OLR_{4ch}$ versus CERES OLR were small at approximately 3 $Wm^{-2}$ and 1 $Wm^{-2}$, respectively. This difference was similar to the trend in the change for the cloudy case in Figure 6b. This kind of trend was seen because several clouds were distributed within the region set by this study, and there were more than five times the number of cloudy cases than cloud-free cases [68]. The difference in the calculated OLR and the CERES OLR was generally similar in partly cloudy cases, but the difference with the CERES OLR was reduced in cases with many clouds, such as mostly cloudy and overcast cases at more than 3 $Wm^{-2}$ and 5 $Wm^{-2}$. In contrast, there was a clear difference in the RMSE of OLR values calculated for cloud-free cases and the CERES OLR values in the southern hemisphere summer. In the northern hemisphere summer (June, July, and August in Figure 6), $OLR_{4ch}$ and CERES OLR had a bias and RMSE of $-0.31$ $Wm^{-2}$ and 6.23 $Wm^{-2}$, respectively, which is an improvement of approximately 1 $Wm^{-2}$ compared to the existing algorithm. However, in the southern hemisphere summer (January, February, and December in Figure 6), the bias and RMSE were $-0.26$ $Wm^{-2}$ and 6.05 $Wm^{-2}$, which was an improvement of more

than 2 Wm$^{-2}$. During the southern hemisphere summer, the bias and RMSE of OLR$_{4ch}$ and CERES OLR for the area around the Australian desert were 0.37 Wm$^{-2}$ and 6.52 Wm$^{-2}$, which was a decrease 7.46 Wm$^{-2}$ and 3.73 Wm$^{-2}$ compared to the difference in the single-channel OLR$_{1ch}$ and CERES OLR. In regions like the Australian desert, which are dry and have a high surface temperature, changes in OLR are sensitive to absorption gas in the atmosphere [10,41–44]; therefore, the OLR calculation algorithm must be built using channel information that can properly reflect these changes.

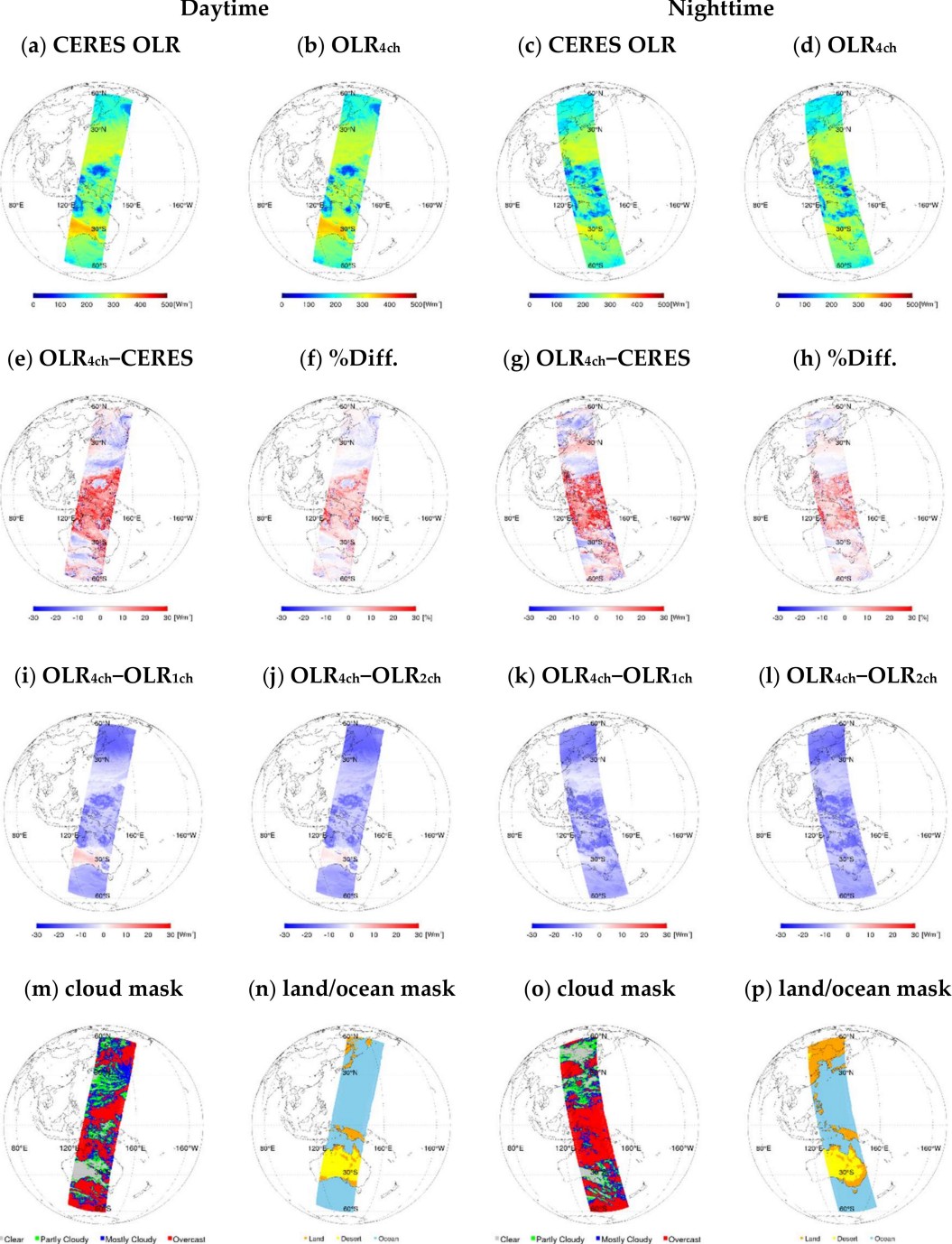

**Figure 5.** CERES OLR and OLR$_{4ch}$ for the daytime (left)/nighttime (right) case of January 4, 2017 (**a,b,c,d**), the difference between these two OLR values (**e,f,g,h**), and the distribution of the differences among OLR$_{1ch}$, OLR$_{2ch}$, and OLR$_{4ch}$ (**i,j,k,l**). The cloud mask and land/ocean mask for each case are shown in figure (**m,n,o,p**). "%Diff." is (OLR$_{4ch}$−CERES)/CERES × 100%.

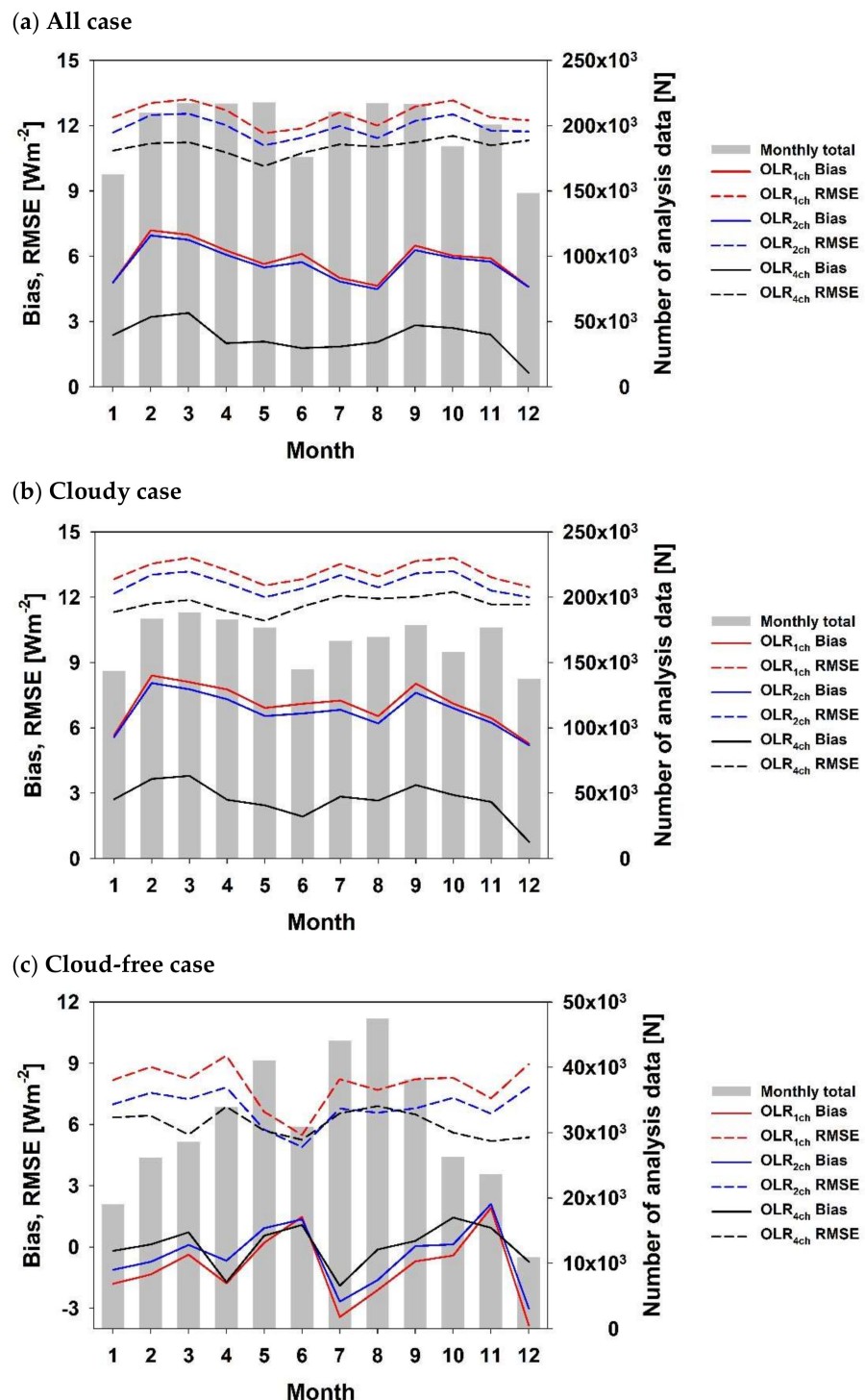

**Figure 6.** Bias and RMSE of calculated AHI OLR and CERES OLR for each monthly case in 2017 and the number of cases analyzed.

**Table 7.** Mean of bias, RMSE, and correlation coefficient (R) for the CERES OLR and AHI OLR calculated for all cases in 2017 and the number of analysis cases. The unit for bias and RMSE is $Wm^{-2}$.

| | | All | Cloudy | | | | Cloud-free | | |
|---|---|---|---|---|---|---|---|---|---|
| | | | Total | Partly | Mostly | Overcast | Total | Ocean | Land (Desert) |
| $OLR_{1ch}$ | Bias | 5.81 | 7.05 | 3.86 | 6.89 | 8.26 | −1.02 | 0.72 | −1.58 (−4.43) |
| | RMSE | 12.51 | 13.18 | 8.94 | 12.34 | 14.70 | 7.95 | 7.16 | 8.22 (8.26) |
| | R | 0.97 | 0.97 | 0.95 | 0.91 | 0.96 | 0.97 | 0.95 | 0.97 (0.94) |
| $OLR_{2ch}$ | Bias | 5.64 | 6.74 | 3.96 | 6.71 | 7.74 | −0.43 | 1.12 | −0.93 (−3.38) |
| | RMSE | 11.91 | 12.62 | 8.21 | 11.76 | 14.18 | 6.84 | 6.21 | 7.07 (6.90) |
| | R | 0.97 | 0.97 | 0.96 | 0.92 | 0.96 | 0.98 | 0.97 | 0.98 (0.95) |
| $OLR_{4ch}$ | Bias | 2.28 | 2.70 | 3.07 | 3.68 | 2.13 | 0.05 | 0.88 | −0.25 (−0.03) |
| | RMSE | 11.03 | 11.69 | 7.57 | 11.15 | 13.06 | 6.02 | 5.51 | 6.22 (6.61) |
| | R | 0.97 | 0.97 | 0.95 | 0.92 | 0.96 | 0.97 | 0.95 | 0.98 (0.95) |
| N | | 2,377,853 | 2,007,514 | 4,019,34 | 482,414 | 1,123,166 | 370,339 | 115,372 | 254,967 (155,604) |

## 5. Summary and Conclusions

This study improved the accuracy of calculating OLR by adding different channel data to the calculation algorithm in Kim et al. [4], which used the Himawari-8 AHI's window and water vapor channels. The OLR results calculated were classified according to cloud and surface characteristics and compared to CERES OLR. One algorithm that uses the window channel, which properly reflects OLR changes according to the cloud and surface characteristics ($OLR_{1ch}$), and another that uses data on water vapor, which is the absorption gas with the largest effect on OLR changes ($OLR_{2ch}$), were developed. However, when geostationary satellite narrowband channel data was used to calculate OLR, the changes in OLR due to clouds and absorption gas were not adequately reflected due to the channel data that was used [4,10,15]. Therefore, in this study, tests were performed on the OLR sensitivity of each channel based on the COT, AOT, and atmospheric concentration of absorption gases, as detailed in Section 3.1. As a result, an OLR calculation algorithm that uses $O_3$ and $CO_2$ channels instead of just window and water vapor channels ($OLR_{4ch}$) was developed and improvements in OLR calculation accuracy were expected. Furthermore, because $CO_2$ has a great effect on global climate change, and $CO_2$ concentrations and increases in radiative forcing due to increases in its concentrations have an important effect on future weather and climate change predictions [69–71], it is desirable to use channel information that is related to this.

The L-F and F-OLR conversion processes given in Sections 3.2 and 3.3, respectively, were performed to calculate $OLR_{4ch}$ using AHI narrowband data. The narrowband irradiance of each channel that was converted in the L-F conversion process showed a %RMSE of less than 0.20% with the narrowband irradiance simulated in the radiative transfer model. The $OLR_{4ch}$ calculated in the F-OLR conversion process showed a %RMSE of 1.87% with the $OLR_{3.3-100}$ simulated in the radiative transfer model. The Garand vertical profile data [65] were used as inputs in the radiative transfer model to perform simulations of various atmospheric conditions. The $OLR_{4ch}$ calculated in this process showed a 2017 yearly averaged bias of 2.28 $Wm^{-2}$ and an RMSE of 11.03 $Wm^{-2}$ with the CERES OLR. This was 3.36 $Wm^{-2}$ and 0.88 $Wm^{-2}$ less than the bias and RMSE of $OLR_{1ch}$, $OLR_{2ch}$, and CERES OLR. This is because the calculation results of $OLR_{4ch}$, which used data from various channels, largely improved upon the differences with CERES OLR in mostly cloudy and overcast areas in the cloudy areas category, and desert areas among cloud-free areas (see Table 7).

OLR has a close relationship with cloud-related rainfall and global warming [72] and has an important role in atmospheric and oceanic circulation [73]. As it is an important factor that determines changes in climate and weather, it must be monitored in real time [74]. OLR can also be calculated in global-scale climate models; however, uncertainty is high for cloud characteristics and weather data used as inputs in these models [75], and the data the latter provide do not have detailed spatiotemporal resolutions [9]. Therefore, their ability to predict regional climate is limited and they include systematic

errors [76]. It is thus very important to use geostationary satellites to produce highly accurate OLR data with a high spatiotemporal resolution. There must be further developments in and improvements to OLR calculation algorithms based on geostationary satellites [4,38].

**Author Contributions:** Conceptualization, B.-Y.K. and K.-T.L.; methodology, B.-Y.K. and K.-T.L.; software, B.-Y.K.; validation, B.-Y.K.; formal analysis, B.-Y.K. and K.-T.L.; investigation, B.-Y.K.; writing—original draft preparation, B.-Y.K.; writing—review and editing, B.-Y.K. and K.-T.L.; visualization, B.-Y.K.; supervision, K.-T.L.

**Funding:** This research was funded by Electronics and Telecommunications Research Institute (ETRI) and National Meteorological Satellite Center (NMSC) of Korea Meteorological Administration (KMA) under Grant No. NMSC-2019-01.

**Acknowledgments:** This work was supported by the "Development of Radiation/Aerosol Algorithms" project, funded by ETRI, which is a subproject of the "Development of Geostationary Meteorological Satellite Ground Segment (grant number: NMSC-2019-01)", a program funded by NMSC (National Meteorological Satellite Center) of KMA (Korea Meteorological Administration).

**Conflicts of Interest:** The authors declare no conflict of interest.

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
