# Peer review of "Using the Himawari-8 AHI Multi-Channel to Improve the Calculation Accuracy of Outgoing Longwave Radiation at the Top of the Atmosphere"

_remotesensing, doi:10.3390/rs11050589_

Round 1

Reviewer 1 Report

This paper delivers an algorithm which improves the accuracy of calculating outgoing longwave radiation (OLR) at the top of the atmosphere. This data is necessary for observing cyclones, weather events such as El Nino, monsoons and typhoons. The algorithm uses data from geostationary satellite data channels. Window, water vapour, O3 and CO2 channels are used to enhance accuracy. The results are compared to previous work by this author and validated against long-term data from a polar satellite source.

This contribution describes the need for outgoing longwave radiation (OLR) at the top of the atmosphere data well. The difference to previous research carried out by this author is elucidated. Overall the research is clearly explained.

I found the work to be of very high standard and am happy to recommend it for publication in its current form.

Author Response

We appreciate the valuable comments of the reviewer. 

This manuscript revised slightly based on other reviewer's comments.

Reviewer 2 Report

Interesting manuscript, well written. Can be published after major revision

The introduction is short, and bibliographically poor. Please add some extra references e.g.: -line 32 /page 1: next to reference [1] you can also add the reference: Chiacchi et al 2015 Journal of Geophysical Research Volume 120, Issue 5, 2015, Pages 1951-1971, Evaluation of the radiation budget with a regional climate model over Europe and inspection of dimming and brightening

page 2 /line62: the sentence "However, to analyze and predict the radiation budget,
weather, and climate change, OLR must react sensitively to absorption gases in the atmosphere in clear sky (cloud-free) areas." is a pure conclusion of the paper Alexandri et al. 2015 Atmospheric Chemistry and Physics,Volume 15, Issue 22, Pages 13195-13216, On the ability of RegCM4 regional climate model to simulate surface solar radiation patterns over Europe: An assessment using satellite-based observations. Please give the reference.

It is not clear the region of the study. Please put the region of interest within a frame (figure 1) or give the lat/lon within the text

SBDART is a radiative transfer model that used in recent studies about solar of terrestial radiation very often. Please enrich the description of the model (e.g write the accuracy etc etc). You can find many details at the paper I suggest above (Alexandri et al., 2015) or the paper Alexandri et al., 2017, Atmospheric ResearchVolume 188, Pages 107-121, A high resolution satellite view of surface solar radiation over the climatically sensitive region of Eastern Mediterranean.

The method followed here is the same with the method followed by -

-Kawamoto 2008, jrl Article number L17809, Relative contributions to surface shortwave irradiance over China: A new index of potential radiative forcing

-Kawamoto et al., 2010 atmos.research,96 (2-3), pp. 337-343, Geographical features of changes in surface shortwave irradiance in East Asia estimated using the potential radiative forcing index

- Alexandri et al., 2015 (given above)

-Alexandri et al., 2017 (given above)

Please add the necessary references into your manuscript.

The section 3 is well written and the functions are clear for the reader

Author Response

We appreciate the valuable opinions and comments of the reviewer, thanks to which we could revise our manuscript to largely enhance its quality. We have made the following revisions in accordance with the additional comments of the reviewer.

1. We added the references as follow (lines 30-33):

“Outgoing longwave radiation at the top of the atmosphere (TOA OLR) is an indicator that can describe the overall state of the earth-atmosphere system [1–4]. Also, OLR is an important radiation budget when balanced with net shortwave radiation at the top of the atmosphere and used in climate studies related to energy balance [5–10].”

2. This sentence means that OLR must change sensitively in clear sky (cloud-free) atmospheric conditions. Therefore, the sentence was revised as follow (lines 63-65):

“However, to analyze and predict the radiation budget, weather, and climate change, OLR must react sensitively to absorption gases in the atmosphere in clear sky (cloud-free) condition.”

3. We added content regarding the performance/accuracy related to SBDART as follow (lines 140-144):

“SBDART has been used in many studies because it has high accuracy with less than 3% error compared to the shortwave/longwave radiation spectrum measured by Atmospheric Emitted Radiance Interferometer (AERI) [60,61], Precision Spectral Pyranometer (PSP), and Normal Incidence Pyrheliometer (NIP) [59], and it calculates various atmospheric conditions very quickly [3,4,38,55,62,63].”

Reviewer 3 Report

Review comments on manuscript “Using the Himawari-8 AHI Multi-Channel to Improve the Calculation Accuracy of Outgoing Longwave Radiation at the Top of the Atmosphere”

Authors: B. Kim, and K. Lee

MS No.: remotesensing-448915

MS Type: Research article

This paper presents the improvements in OLR calculations with Himawari-8 observations when multi-channels are used. The four channels used include a window channel, a water vapor channel, CO2 and O3 channel. The paper provides new progress in OLR retrievals and is of interest to the community. I recommend publication after some minor revisions.

Comments:

1)    OLR is sensitive to cloud properties, as the authors showed. Among the four channels used for the retrieval, I can see that cloud height info is included (the window channel), but how is the COT taken into account? I don’t see any of the four channels contain COT information.

2)    The properties of the channels used should be introduced upfront.

3)     P1 Line 22: “improve” change to “mitigated”.

Author Response

We appreciate the valuable opinions and comments of the reviewer, thanks to which we could revise our manuscript to largely enhance its quality. We have made the following revisions in accordance with the additional comments of the reviewer.

1. As commented by the reviewer, OLR is very sensitive to cloud characteristics (such as cloud height and COT). The observed AHI channel data have already provided radiance data reflecting cloud characteristics. Therefore, we do not use COT directly to calculate OLR (the same principle applies to other meteorological variables). In Section 3.1, COT was used as input for the SBDART model in the sensitivity test according to the cloud characteristics (COT).

2. We added a table (Table 1) for summarizing the channel-specific characteristics of the Himawari-8 AHI used in this study (line 88).

3. We revised the word “improved” to “mitigated” (line 22).

Round 2

Reviewer 2 Report

accept as is